# Design and Preparation of Localized Heat-Resistant Coating

**DOI:** 10.3390/polym14153032

**Published:** 2022-07-27

**Authors:** Zaiming Lin, Yihan Chen, Zhuang Ma, Lihong Gao, Wenhua Chen, Guohua Chen, Chen Ma

**Affiliations:** 1School of Materials Science and Engineering, Huaqiao University, Xiamen 361021, China; linzm@stu.hqu.edu.cn (Z.L.); chenyihan19990521@163.com (Y.C.); scucwh@163.com (W.C.); hdcgh@hqu.edu.cn (G.C.); 2School of Material Science and Engineering, Beijing Institute of Technology, Beijing 100081, China; hstrong929@bit.edu.cn (Z.M.); gaolihong@bit.edu.cn (L.G.)

**Keywords:** composite coatings, localized heat, reduced graphene oxide, high thermal conductivity

## Abstract

Localized heat sources, such as flame guns and high-energy lasers, can cause severe damage to conventional materials. In this study, a novel localized heat-resistant coating with a high in-plane thermal conductivity was designed and prepared. Reduced graphene oxide (rGO) effectively improved the in-plane thermal conductivity of the polyvinyl alcohol (PVA) film, while maintaining the thermal insulation of the resin matrix in the through-plane direction. This characteristic of the rGO/PVA film was combined with the thermal insulation of boron-modified phenolic resin (BPF), and the prepared composite coating with two layers of rGO/PVA films effectively lowered the back-surface temperature in the flame ablation test from 151 to 107 °C. In addition, the area of the ablation-affected region of coating was increased to 103.6 cm^2^ from 31.9 cm^2^, indicating an excellent heat transfer performance. The layer-by-layer structure could realize the compatibility of high in-plane thermal conductivity and good through-plane thermal insulation. The synergy of these two different characteristics is demonstrated to be the key to improving the localized heat-resistant performance of the composite coating. This study effectively expands the application range of high-conductive film, and the obtained coating could act as a shield against butane flame, high energy lasers, and other localized heat.

## 1. Introduction

Localized heat is a large energy input inside a small region, such as is produced by a flame gun [1] or high-energy laser [2,3,4]. Owing to the high energy density, many conventional materials can be destroyed in a short time [5,6]. Thermal protection systems effectively prevent thermal attacks, and are widely used in aerospace applications [7,8]. However, such materials always have high thicknesses and weak bonds with substrates [9,10]. The prevention of localized heat using a relatively simple method while maintaining the original performance of conventional metal substrates remains a challenge.

Protective coatings have successfully been prepared on the surfaces of metal substrates. Such coatings exhibit special characteristics depending on the features of the heat source. For high-energy lasers, researchers have designed metal or ceramic coatings with high reflectivity. High reflectivity could effectively consume a high ratio of laser energy and lower the heat deposition [11,12]. However, this high-reflection characteristic cannot prevent localized flame ablation. Thermal insulation prevents flame ablation, but low thermal conduction concentrates the thermal energy inside the ablated region, leading to an ultra-high temperature and the failure of the protective coating [13]. Thus, a single thermal insulation characteristic of the coating can hardly meet the thermal protection requirement. However, if the thermal conduction in the in-plane direction of the coating can be improved based on the through-plane thermal insulation, coatings with such anisotropic thermal conductive characteristics could potentially be applied for localized heat-resistant coatings [14,15].

To determine the anisotropic thermal conductivity during thermal ablation, components with low and high thermal conductivities should be identified. Boron phenolic resin (BPF) is a modified product with boron atoms introduced into phenolic resin [16,17]. This novel resin shows excellent thermal stability and has a high residual char yield because the bond dissociation enthalpy for B–O bonds is higher than that for C–O bonds [18]. Thus, BPF has been applied in ablation heat-resistance research fields, and many researchers are devotedly studying the thermal decomposition mechanism and further improving its performance [19,20]. In addition, polymer matrix composites or composite coatings have been proven to have relatively good mechanical properties [21,22,23], which are also beneficial for thermal protection. The thermal insulation of a BPF coating can prevent heat deposition on the surface by transferring the heat to the bottom of materials; however, it can also result in an extremely localized temperature field [24]. The thermal concentration issue must be addressed to use BPF as a matrix to design a localized heat-resistant coating.

Graphene is known as an excellent filling material owing to its theoretical thermal conductivity (larger than 4000 W/m∙K) [25]. Many types of films with high thermal conductivities have been designed and fabricated based on horizontally distributed graphene particles [26,27,28,29]. These thermally conductive films effectively transfer the localized heat from integrated circuits to the surroundings [30]. However, because of the poor thermos ability of the resin matrix, such films cannot withstand the high temperatures of hundreds of degrees. Nevertheless, we are interested in horizontally distributed graphene particles that retain high thermal conductivity even upon the pyrolysis of the resin matrix. Therefore, the thermally conductive films can be potentially used as modified components in the localized heat-resistant coatings.

In this study, based on the good film forming property of polyvinyl alcohol (PVA) and flexibility of the PVA film [31], reduced graphene oxide (rGO)/PVA composite films with different rGO contents are prepared. Both the in-plane and through-plane thermal conductivities of the films are measured and compared. Based on their results, a composite coating with a high in-plane thermal conductivity and a low through-plane thermal conductivity is designed and prepared. A thermal ablation test is conducted on the surface of the coating, and the back-surface temperature is monitored to evaluate the thermal protection performance. A film comprising only BPF was used as control to elucidate the role of rGO/PVA. Additionally, the ablation-affected area of the ablated samples is assessed and compared to understand the improvement in the heat transfer performance in the in-plane direction.

## 2. Materials and Methods

### 2.1. Materials

Graphite powder (with a size of 8000 mesh, a density of 2.25 g/cm^3^, and a purity of 99.5%) was purchased from the Xiamen Kenna Graphene Technology Co., Ltd. (Xiamen, China). Potassium permanganate (with a CAS of 7722-64-7) was purchased from the Xilong Science Co., Ltd. (Guangzhou, China). Concentrated sulfuric acid (with a CAS of 7664-93-9), L-ascorbic acid (with a CAS of 50-81-7), and glycerol (with a CAS of 56-81-5) were provided by the Sinopharm Group Chemical Reagent Co., Ltd. (Shanghai, China). Additionally, PVA (of model 1788, with a melting point of 200 °C, a density of 1.3 g/cm^3^, and a degree of alcoholysis of 87.0–89.0%) was purchased from the Shanghai Aladdin Biochemical Technology Co., Ltd. (Shanghai, China). Resol-type BPF was provided by the Bengbu Tianyu High Temperature Resin Materials Co., Ltd. (Bengbu, China). (of model FB90, with a pyrolysis temperature of 527 °C and a carbon yield at 900 °C of 70%).

### 2.2. Fabrication of PVA/rGO Film

The GO particles used in this study were prepared using the Hummers method [32]. Blade-coating was used to fabricate the PVA/rGO films using the prepared GO particles. Subsequently, 36 g PVA was weighed and slowly added to deionized water (360 mL, 20 °C), sealed, and magnetically stirred for 2 h until the solution was clear (i.e., there were no granular suspension). Then, 4 g glycerol was added, and the solution was heated to 100 °C in a water bath for 2 h, and a PVA solution with a mass fraction of 9% was obtained.

An electronic moisture tester was used to measure the solid content of the GO solution (e.g., 1.25% solid content). The GO solution (40 mL) was ultrasonicated for 2 min using an ultrasonic multifunctional testing machine. Furthermore, 100 mL of the PVA solution was added and the mixture was ultrasonicated for 2 min. After the solution was cooled to 20 °C, 3 g L-ascorbic acid (mass ratio of reductant to GO, 6:1) was added and the mixture was magnetically stirred at room temperature for 2 h to yield a 5% rGO/PVA solution. An rGO/PVA film was fabricated using a coating machine. The thickness of the film was precisely controlled to be 0.1 mm by adjusting the gap between the scraper and smooth steel plate. The film was placed in a vacuum-drying oven and maintained at 80 °C for 7 h. After stripping off the film from the steel plate, an rGO/PVA film with a mass fraction of 5% was obtained. Similarly, rGO/PVA films with mass fractions of 7% and 9% were prepared by changing the additive amount of rGO.

### 2.3. Fabrication of Localized Heat-Resistant Coating

The localized heat-resistant coating consisted of BPF and rGO/PVA layers. Based on the thicknesses of the rGO/PVA films (0.1 mm) and the final coating (2.5 mm), the thickness of the BPF layer was calculated according to the number of rGO/PVA layers. The BPF was dissolved in alcohol and coated on an aluminum substrate using a blade-coating method, and the rGO/PVA film was pasted onto the uncured BPF coating. The coating was then cured for 4 h in a convection oven at 60 °C. This procedure was repeated multiple times, and only BPF was coated in the final round. The coating was again heated in a convection oven for 6 h at 60 °C, 2 h at 100 °C, 2 h at 120 °C, and 2 h at 150 °C for complete polymerization. The obtained samples with two and three layers of rGO/PVA films were denoted as the BPF-2GP and BPF-3GP coatings, respectively. To study the role of the rGO/PVA films, a control sample containing only BPF was prepared and denoted as BPF-N. The steps in the preparation procedure of the localized heat-resistant coating are shown in Figure 1.

### 2.4. Characterization and Testing

Scanning electron microscopy (HITACHI S4800, Tokyo, Japan, SEM) was used to examine the micro-morphologies of the samples before and after the thermal ablation test (accelerating voltage: 3 kV). Thermogravimetry analysis (NETZSCH STA 449 F3, Zurich, Switzerland, TGA) was used to detect the residual char yield of different films. A laser thermal instrument (Nanoflash LFA 447, Selb, Germany) was used to measure thermal diffusivity (*α*), and differential scanning calorimetry (Mettler Toledo, Zurich, Switzerland, DSC) was used to measure the heat capacity (*Cp*) of samples. Based on the density of the sample (*ρ*), the thermal conductivity (*k*) was calculated using the following equation:*k* = *α* × *ρ* × *Cp*

A thermal ablation platform was built to study the thermal protection performance of the composite coatings. As shown in Figure 2, the samples are fixed on a metal rack and a butane flame gun is used as the heat source. To obtain a relatively stable ablation state, the distance between the sample and muzzle of the butane flame gun was set to 10 cm and the ablation time was set to 30 s. The temperature of the surface of the sample, which was tested using a thermocouple, was approximately 1000 °C. In addition, an infrared thermometer was placed behind the sample to monitor the back-surface temperature in real-time. After the ablation test, the ablated area was accurately calculated to evaluate the anisotropic thermal conductivity.

## 3. Results

### 3.1. Characteristics of the rGO/PVA Film

With the prepared few-layer GO particles, the rGO/PVA film with rGO mass fractions of 5% was obtained using the aforementioned preparation method. The film was uniformly black (Figure 3a) and flexible (Figure 3b). It can be easily crimped and reverted to its original shape. This outstanding flexibility made it easy to use the film as a modification filler in composite coatings, which changed the shape of the coating substrate to be relatively complex. In addition, the surface of the film is flat, and the thickness is very thin. This guarantees that heat transfer only occurs in the in-plane direction, which is the key point for localized heat protection.

Figure 3c,d show the surface and the cross-section morphologies of the rGO/PVA film, respectively. A compact and smooth surface was observed, guaranteeing the uniform performance of the entire film. With the optimization of the coating and drying processes, no holes were observed in the rGO/PVA film. According to the cross-sectional morphologies, there was no agglomeration, indicating that the ultrasonic dispersion effectively dispersed rGO in the PVA solution. The uniformly dispersed rGO nanoparticles and the flexible characteristic could build the heat transferring pathway on any shape of substrate, which lays the foundation for the composite coating. The FTIR spectra of pure PVA and rGO/PVA films containing 5 wt.%, 7 wt.% and 9 wt.% rGO is shown in Figure 3e. The strong hydroxyl band for free and hydrogen bonded alcohols can be observed in the range of 3100–3500 cm^−1^ [33]. It is noteworthy that this band of rGO/PVA films show a shift to a lower wave number compared with the band of pure PVA. This indicates the dissociation of the hydrogen bonding among the hydroxyl groups [34], which further proves that hydrogen bonding could be the main chemical interaction between PVA and rGO.

The addition of rGO particles significantly affected the residual char yield of the rGO/PVA film (Figure 4a). As the mass fraction of rGO increased from 5% to 9%, the residual char yield at 1000 °C of the composite film increased from 2.6% to 14.7%. The significant improvement in the residual char yield had a positive effect on the thermal protection performance of the films. Pure PVA exhibited low thermal conductivity [35], which was similar to the conventional resin matrix. However, with the modification of rGO, the thermal conductivity of the composite films significantly improved (Figure 4b). When the rGO content was 5%, the in-plane thermal conductivity was 0.73 W/m∙K. When the rGO content was 9%, the thermal conductivity increased to 2.02 W/m∙K. As GO particles were distributed layer by layer in the composite film, the overlap between different particles was the key to improving the thermal conductivity. However, there was limited overlap in the in-plane direction, unlike the full contact in the through-plane direction [36]. Thus, the through-plane thermal conductivity was always lower than the in-plane thermal conductivity. When the rGO content was 9%, the through-plane thermal conductivity was only 0.52 W/m∙K, which was only a quarter of the in-plane thermal conductivity at the same concentration. The unique anisotropic thermal conductivity promoted different heat transfer performances in different directions, which was desirable in the thermal protection research field. According to these results, the 9% mass fraction rGO/PVA film exhibited superior thermal stability and thermal conductivity. Although the increase in rGO mass fraction could improve the thermal conductivity of the film, it can also result in the problems of reuniting and poor flexibility. In our study, 9% mass fraction has been proven to be the maximum figure maintaining the basic film properties. Therefore, the rGO/PVA film with 9% mass fraction was chosen to prepare a localized heat-resistant coating.

### 3.2. Morphologies and Thermal Protection Performance of the Localized Heat-Resistant Coating

A pure BPF layer and two layers of the rGO/PVA films with an rGO mass fraction of 9% are combined to prepare a localized heat-resistant coating, namely BPF-2GP (Figure 5a). Pure BPF was coated on the top surface of the coating, which could prevent the surface directly breaking through when facing thermal ablation. According to the cross-section of the composite coatings (Figure 5b), the BPF and rGO/PVA layers can form an orderly arrangement layer-by-layer. Such a structure optimally benefits form the characteristics of each component functional layer [14,26]. In addition, the composite coating is compact, and no holes or cracks can be observed inside the coating, particularly at the interface between the BPF and rGO/PVA layers (Figure 5c). The excellent combination of these two layers results in the strong adhesion strength of the composite coating, which avoids the formation of cracks inside the coating, particularly at high temperatures. Furthermore, the different rGO/PVA layers could be separated by the BPF layer, avoiding the thermal conduction in the through-plane direction [37]. The layer distribution could preliminarily understand the heat transfer process.

The layer-by-layer structure is conducive to high thermal protection [38]. Figure 6 shows the morphologies of samples that are thermally ablated using a butane gun for 30 s at a distance of 10 cm. For the pure aluminum alloy substrate with a 1 mm thickness (Figure 6a), a large molten hole appears in the ablated region, indicating the failure of the sample. When the aluminum substrate was coated with BPF-N coating, although the coating shows a decomposition change, the substrate is not destroyed (Figure 6b), indicating that BPF-N exhibits some thermal protection performance. During thermal ablation, the high temperature results in the decomposition of BPF and the formation of residual char. The decomposition process could effectively consume a large amount of energy, which has a positive effect on the protection performance [39,40]. In addition, the formed residual char owns porous structure that has been proved to be the key for the heat insulation property [41]. According to the back-surface temperature (Figure 6e), the highest temperature is 151 °C. Although this temperature is lower than the melting point of the aluminum alloy substrate, an ablation crater appears at the center of the ablated region. Although the thermal insulation performance of BPF is outstanding, it exacerbates heat concentration, which leads to a gradual decline in the thermal protection performance [42].

The ablated region of BPF-2GP is smoother, and no ablation craters are observed (Figure 6c). Additionally, the ablated region does not reach the edge of BPF-N. However, the ablated regions of BPF-2GP cover the entire sample. Therefore, the addition of a rGO/PVA layer could improve the heat transfer rate during thermal ablation. The improved heat transfer performance in the in-plane direction effectively mitigates the ultra-high temperature in the center of the ablated region. This could be the key factor to avoid the formation of the ablation crater.

Due to the addition of the rGO/PVA layer, the highest back-surface temperature in the center of ablated region decreases from 151 to 134 °C, indicating the improvement of the protection performance [43]. In addition, the highest back-surface temperature further decreases to 107 °C when the number of rGO/PVA layers increases to three (BPF-3GP). This is especially the case with BPF-2GP and BPF-3GP, which show a lower rate of temperature rise in the initial 5–15 s. This indicates that a rGO/PVA layer play a significant role in the initial ablation process. The back-surface temperature is the main parameter evaluating the protection performance. Therefore, it can be demonstrated that the introduction of rGO/PVA layers does improve the thermal protection performance of the composite coating. Additionally, with the increase in the number of rGO/PVA layers, the protection performance also shows a further improvement. As the thickness of the composite coating has been fixed to 2.5 mm, the coating could hardly maintain the separation of different rGO/PVA films if four layer of films were introduced, which could further result in the failure of the thermal insulation in the through-plane direction. Therefore, three layer of rGO/PVA films are considered to be the most appropriate in this study. By comparing the morphologies of the ablated sample, both BPF-2GP and BPF-3GP show a smoother surface than BPF-N. The in-plane thermal conduction could be a significant path for accelerating/heat transfer from the center to the surroundings.

To further verify the in-plane thermal transfer, BPF-N and BPF-3GP samples of dimensions 150 mm × 150 mm are prepared. The comparison of thermal conduction performances is emphasized owing to an increase in the sample size. Figure 7a shows the ablated morphology of BPF-N. The maximum size of the ablation-affected region is 69.3 mm. Additionally, the area of this region is calculated using image processing [14]. Photoshop is used to accurately subdivide the ablation-affected and unaffected regions (Figure 7b). The area of the ablation-affected region is calculated by comparing the number of pixels in the region to that of the entire sample, which has an area of 225 cm^2^. The ablation-affected region of BPF-N is 31.9 cm^2^. With the modification of the rGO/PVA layers, the ablation-affected region of BPF-3GP increases to 103.6 cm^2^, which is 3.2 times larger than that of BPF-N. The increase in the ablation-affected region can directly prove the improvement of the heat transferring performance. The high heat transferring rate promotes the equal distribution of the deposited energy, which could effectively solve the heat concentration problem in the ablated region [44]. Thus, BPF-3GP composite coating shows a much better protection performance than pure BPF coating.

These results indicate that the prepared composite coatings can effectively protect substrates from localized heat attacks. The rGO/PVA film realizes the rapid heat transfer in a very thin thickness, which has been proven to be the key to relieving the ultrahigh temperature field in the ablated region and to avoid the formation of the ablation crater. The decomposition of BPF can consume the energy, and the porous residual char that is the pyrolysis product of BPF can prevent the heat transfer in the through-plane direction. The combination of the rGO/PVA film and the BPF layer achieves the synergy between these two different characteristics. As a result, when the deposited heat transfers from the surface to the substrate of the composite coating, the heat can be isolated in the through-plane direction and transferred to the surroundings repeatedly. Therefore, the composite coatings exhibit good protection performance, and can act as the shield against the butane flame used in this study, high energy laser, and other kinds of localized heat.

## 4. Conclusions

A composite coating with high in-plane thermal conductivity and low through-plane thermal conductivity was prepared. Ablation resistance performance was studied using a thermal ablation test. The obtained rGO/PVA film exhibited anisotropic thermal conductivity. The thermal conductivity was observed to be 2.02 W/m∙K, which significantly promotes the heat transfer rate of the composite coating when facing localized thermal damage. The back-surface temperature of the composite coatings, comprising the two layers of rGO/PVA films, decreased from 151 to 107 °C when it was ablated for 30 s. The area of the ablation-affected region increased from 31.9 to 103.6 cm^2^, which indicated the transfer of heat from the localized heat region to the surroundings. It has been demonstrated that the composite coating can effectively protect from the localized heat. Additionally, the synergy of high in-plane thermal conductivity and good through-plane thermal insulation is the key to achieve the excellent protection performance.

## Figures and Tables

**Figure 1 polymers-14-03032-f001:**
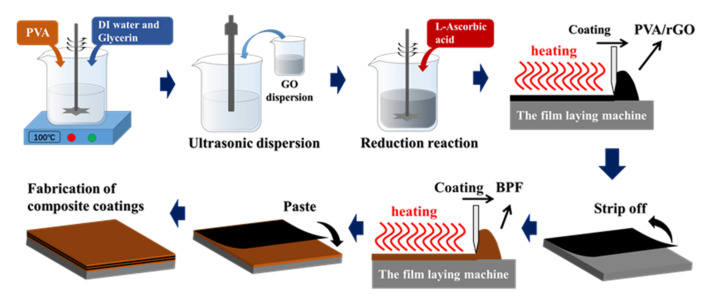
Preparation procedure of localized heat-resistant coatings.

**Figure 2 polymers-14-03032-f002:**
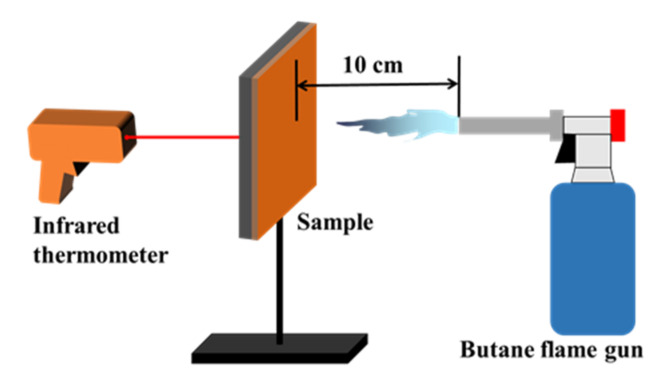
Schematic depiction of the thermal ablation platform.

**Figure 3 polymers-14-03032-f003:**
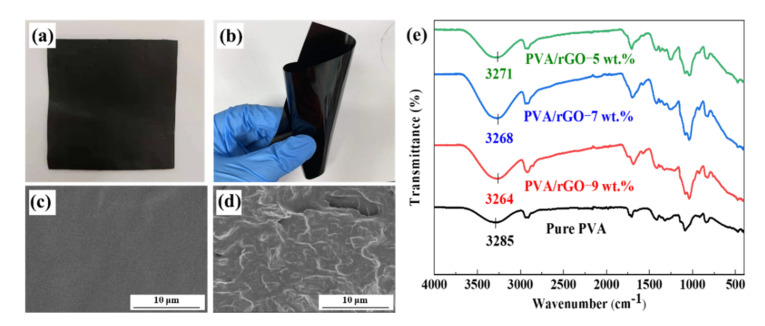
(**a**,**b**) Macro-morphologies of the rGO/PVA film with rGO mass fractions of 5%; (**c**) micro-morphologies of the surface of the rGO/PVA film; (**d**) micro-morphologies of the cross-section of the rGO/PVA film; (**e**) FTIR spectra of pure PVA and rGO/PVA films containing 5 wt.%, 7 wt.% and 9 wt.% rGO.

**Figure 4 polymers-14-03032-f004:**
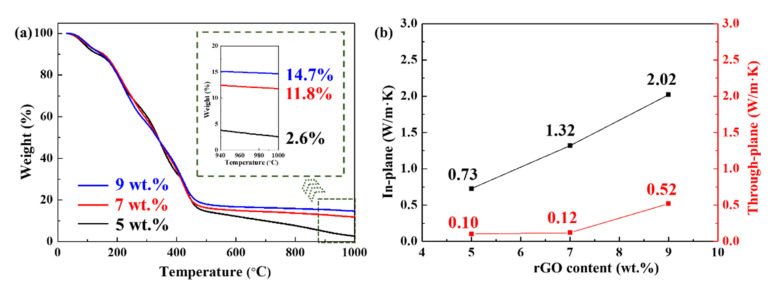
Char yield and the thermal conductivity of rGO/PVA films with the rGO mass fractions of 5%, 7%, and 9%, respectively. (**a**) TGA thermograms; (**b**) thermal conductivity in different directions.

**Figure 5 polymers-14-03032-f005:**
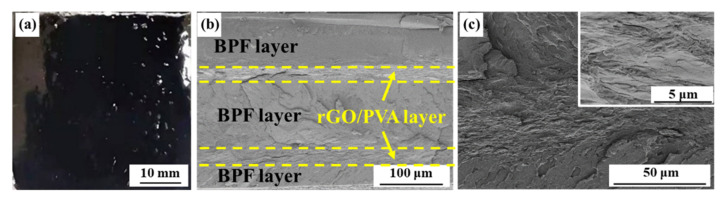
Morphologies of BPF-2GP. (**a**) Surface of the coating; (**b**) cross-section of the coating; (**c**) high magnification of the interface between the BPF and rGO/PVA layers.

**Figure 6 polymers-14-03032-f006:**
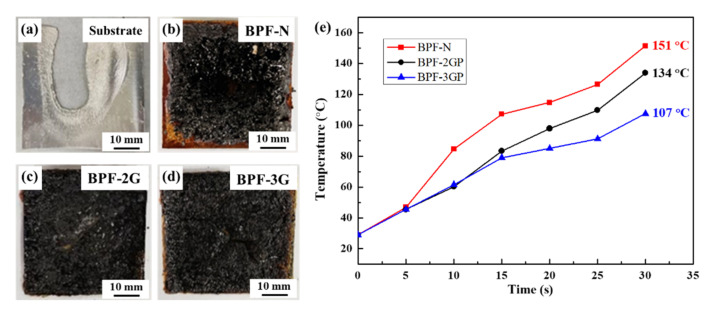
Ablated morphologies and back-surface temperatures of different samples. Photographs of (**a**) pure aluminum alloy; (**b**) BPF-N; (**c**) BPF-2GP; (**d**) BPF-3GP; (**e**) Back-surface temperature over time.

**Figure 7 polymers-14-03032-f007:**
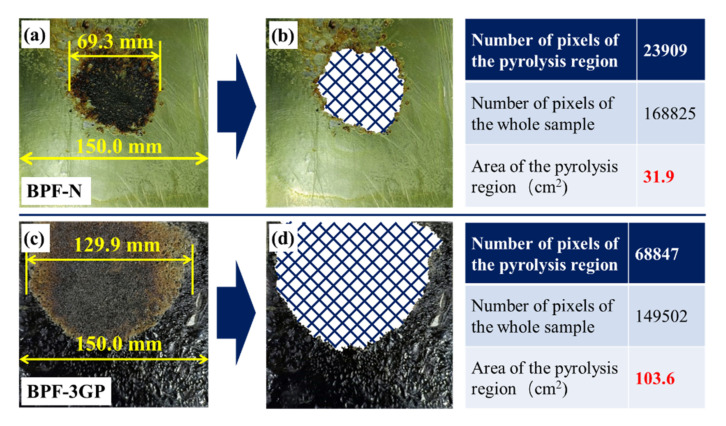
Calculation of the ablation-affected area. (**a**) Morphology of ablated BPF-N; (**b**) subdivision of the ablated region and the calculation of pixels; (**c**) Morphology of the ablated BPF-3GP sample; (**d**) subdivision of the ablated region and the calculation of pixels.

## Data Availability

Not applicable.

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
