# Peer review of "Design and Preparation of Localized Heat-Resistant Coating"

_polymers, 2022, doi:10.3390/polym14153032_

Round 1

Reviewer 1 Report

The author need to be considered the following points to improve the quality of this manuscript:

1. PVA is not a widely used polymer, please explain the advantage of using the PVA as a base coating material.

2. Please mention clearly the amount of graphene content (wt %) you used, how about the optimum content/

3. The chemical interaction between PVA and graphene must be characterized by FT-IR analysis.

Reviewer 2 Report

The paper seeks to introduce an approach ‘’ Design and preparation of the localized heat-resistant coating’’ However, the authors should consider improving upon the quality to further highlight and emphasis.

1.    Based on the understanding of what should constitute an abstract, consider adding one or two lines highlighting the significance of the study at the end of the abstract.

2.    The introduction needs to be improved by relating to the mechanics of the studied materials and their mechanical characteristics. The references to be included are: 10.1007/s10853-022-06994-3, 10.1016/j.polymertesting.2017.09.009, 10.1016/j.compstruct.2021.114698, 10.1016/j.jiec.2022.06.023 and 10.3390/polym14132662.

3.    Tabulate the materials used in this study with their physical and chemical properties.

4.    Put a space between each variable and its corresponding unit including the percentage sign.

5.    How was the coating thickness calculated?

6.    State the accelerating voltage, working range, and the scale bar used for the SEM analysis.

7.    How was the coating done? Explain in detail how the coating was achieved.

8.    Increase the font size of figure 4. Its hard to read what is written in the figure inside figure 4a. consider making the font sizes of all the figure uniform.

Round 2

Reviewer 1 Report

The manuscripts has been revised accordingly. Please add references in mentioned explanation.

Author Response

Thank you for the reviewer’s good suggestions. We have added the suggested references in the mentioned explanation. The corrected content is shown below.

In this study, based on the good film forming property of polyvinyl alcohol (PVA) and flexibility of the PVA film [31], reduced graphene oxide (rGO)/ PVA composite films with different rGO contents are prepared.

The strong hydroxyl band for free and hydrogen bonded alcohols can be observed in the range of 3100-3500 cm-1 [33].

References

[31] Dmitrenko, M.; Penkova, A.; Kuzminova, A.; Missyul, A.; Ermakov, S.; Roizard, D. Development and characterization of new pervaporation PVA membranes for the dehydration using bulk and surface modifications. Polymers. 2018, 10, 571.

[33] Lim, M.; Kwon, H.; Kim, D.; Seo, J.; Han, H.; Khan, S.B. Highly-enhanced water resistant and oxygen barrier properties of cross-linked poly(vinyl alcohol) hybrid films for packaging applications. Prog. Org. Coat. 2015, 85, 68-75.